# Riemannian Black Box Variational Inference

**Mykola Lukashchuk**[1]     **Wouter W. L. Nuijten**[1]     **Dmitry Bagaev**[1,2]

**İsmail Şenöz**[1,2]      **Bert de Vries**[1,2,3]

[1]Eindhoven University of Technology     [2]Lazy Dynamics     [3]GN Hearing

Eindhoven, The Netherlands

{m.lukashchuk, w.w.l.nuijten, d.v.bagaev, i.senoz, bert.de.vries}@tue.nl

## Abstract

We introduce Riemannian Black Box Variational Inference (RBBVI) for scenarios lacking gradient information of the model with respect to its parameters. Our method constrains posterior marginals to exponential families, optimizing variational free energy using Riemannian geometry and gradients of the log-partition function. It excels with black-box or nondifferentiable models, where popular methods fail. We demonstrate efficacy by inferring parameters from the SIR model and tuning neural network learning rates. The results show competitive performance with gradient-based (NUTS) and gradient-free (Latent Slice Sampling) methods, achieving better coverage and matching Bayesian optimization with fewer evaluations. RBBVI extends variational inference to settings where model gradients are unavailable, improving efficiency and flexibility for real-world applications.

## 1 Introduction

Bayesian Inference focuses on updating beliefs about hypotheses based on newly available evidence. Previously too costly, it is now more feasible thanks to Markov-Chain Monte Carlo (MCMC) algorithms that can sample from difficult posterior distributions and Variational Inference methods that can directly optimize an approximate posterior distribution. However, widely used MCMC methods, such as the No-U-Turn Sampler (NUTS) [12] and well-known variational inference methods such as Automatic Differentiation Variational Inference (ADVI) [17] or Stochastic Variational Inference (SVI) [11] require that the gradient of the log-probability density function with respect to the parameters be computable.

Gradient-free methods, such as Ensemble Slice Sampling (ESS) [14], propose a way to sample from the posterior distribution over parameters even when the gradient is unavailable. Although these methods are gradient-free and nonparametric, they require significant evaluations of the log-probability density function, which may be computationally expensive.

This paper introduces Riemannian Black Box Variational Inference (RBBVI) for approximating posterior marginal distributions, addressing computational challenges in two ways: by constraining the posterior to a predefined exponential family [13], we balance approximation accuracy with computational cost; optimizing variational free energy allows for early stopping, potentially reducing function evaluations compared to sampling methods.

Workshop on Bayesian Decision-making and Uncertainty, 38th Conference on Neural Information Processing Systems (NeurIPS 2024).

RBBVI is particularly well-suited for approximate inference in probabilistic models with black-box components. This makes the proposed method ideal for scenarios where nondifferentiable and computationally expensive operations are frequent or for hyperparameter tuning, where gradients might not be easily accessible. To demonstrate this, we address the problem of inferring the optimal learning rate of gradient descent while training a neural network. In this situation, the gradient of the loss function with respect to the learning rate is not directly available. Since training a neural network is costly, it is crucial to minimize the number of executions during hyperparameter tuning.

## 2 Riemannian Variational Inference

We are concerned with Bayesian inference in a generative model $p(\theta, y)$ with $\theta$ unobserved parameters and $y$ observed data. We are interested in the inference task of determining $p(\theta|y)$. In this paper, we assume a factorizable generative model with independent priors over $(\theta_1, \ldots, \theta_n)$

$$p(y, \theta_1, \ldots, \theta_n) = p(y|\theta_1, \ldots, \theta_n) \prod_i p_i(\theta_i), \tag{1}$$

where $p(y, |\theta_1, \ldots, \theta_n)$ is a generative process that is possibly not differentiable with respect to $\theta$.

Variational inference addresses this by introducing an approximate posterior distribution $q$ on which we want to maximize the evidence lower bound (ELBO) or equivalently minimize the Free Energy

$$\mathrm{F}[q, p](y) = \int q(\theta) \log \frac{q(\theta)}{p(y, \theta)} \mathrm{d}\theta, \tag{2a}$$

where the goal is to compute

$$q^* = \operatorname{argmin} F[q, p](y), \tag{2b}$$

for a given observation $y$.

The problem (2b) is a nonparametric one. To cast it into a parametric optimization problem, we assume the following functional form of $q(\theta_1, \ldots, \theta_n)$ to be a factorized posterior $\prod_i q_i(\theta_i)$, where each constrained $q_i(\theta_i)$ to be a known exponential family

$$q_{\lambda_i}(\theta_i) = \exp(\lambda_i^T T_i(\theta_i) - A_i(\lambda_i) + \kappa_i(\theta_i)), \tag{3}$$

with natural parameters $\lambda_i$, sufficient statistics $T_i$, a base measure $\kappa_i$, and known log-partition function $A_i$. This turns the problem (2b) into the following parametric problem

$$\lambda^* = \operatorname*{argmin}_{\lambda=(\lambda_i, \ldots, \lambda_n)} \mathrm{F}\left[\prod_i q_{\lambda_i}(\theta_i), p\right](y). \tag{4}$$

The fruitful idea is to apply gradient-based methods to the Free Energy objective $F$ with respect to the parameters $\lambda$. This approach was first proposed by Amari [2] with the following gradient update

$$\lambda^{k+1} = \lambda^k - \mathcal{F}^{-1}(\lambda^k)\partial_\lambda F|_{\lambda=\lambda^k}, \tag{5}$$

where $\mathcal{F}$ is the Fisher information matrix. The update rule (5) has gained further popularity, and Khan [15] has developed extensions to the rule. The typical challenge in applying the rule (5) is that $\lambda$ usually lies within an open constraint set, so the update may not always meet the constraints. Lin [20] addresses this issue for positive-definite constraints (e.g., Gamma, Gaussian) using Riemannian gradient descent. It is crucial to note that even when the gradient of $p(y, \theta_1, \ldots, \theta_n)$ with respect to $(\theta_1, \ldots, \theta_n)$ does not exist or is intractable, the gradient $\partial_\lambda F$ can still exist and be tractable, depending on the specific choice of $q$.

Similarly to Lin's approach, we assume that each $\lambda_i$ belongs to a constraint set $\Lambda_i$ that forms a Riemannian manifold with $\mathcal{F}_i$ as its metric. For readers interested in a comprehensive treatment of these concepts, we recommend the book by Absil et al. [1]. For readers new to manifold theory, a manifold $\mathcal{M}$ can be thought of as a set that, at any point $x \in \mathcal{M}$ can be approximated by a vector space. This vector space is called the tangent space $T_x\mathcal{M}$. An important detail is that the dimensionality of $T_x\mathcal{M}$ remains constant for each point $x$ in the manifold. A Riemannian manifold is equipped with a Riemannian metric, which defines a smooth inner product on each tangent space, allowing us to measure distances and angles between tangent vectors. The Riemannian gradient $\partial f(x)$ at a point $x$ on $\mathcal{M}$ is the unique tangent vector in $T_x\mathcal{M}$ that satisfies $\langle \partial f(x), \xi \rangle_x = Df(x)[\xi]$ for all $\xi \in T_x\mathcal{M}$, where $\langle \cdot, \cdot \rangle_x$ denotes the Riemannian metric in $x$. To move along the manifold in

the direction of a tangent vector, we use a retraction $R_x : T_x\mathcal{M} \to \mathcal{M}$. Specifically, for a tangent vector $\xi \in T_x\mathcal{M}$, the curve $\gamma(t) = R_x(t\xi)$ satisfies $\gamma(0) = x$ and $\dot{\gamma}(0) = \xi$.

We equip each $\Lambda_i$ with a Riemannian metric $\mathcal{F}_i$ and a known closed-form retraction $R_i$. We treat $\Lambda = \bigotimes_i \Lambda_i$ as a product manifold with the product retraction $R_\lambda(\xi) = (R^1_{\lambda_1}(\xi_i), \ldots, R^n_{\lambda_n}(\xi_n))$, leading to the following update rule

$$\lambda^{k+1} = R_{\lambda^k}\left(-\mathcal{F}^{-1}(\lambda^k)\partial_\lambda F|_{\lambda=\lambda^k}\right), \tag{6a}$$

where

$$\mathcal{F}(\lambda) = \begin{bmatrix} \mathcal{F}_1(\lambda_1) & \mathbf{0} & \mathbf{0} \\ \mathbf{0} & \ddots & \mathbf{0} \\ \mathbf{0} & \mathbf{0} & \mathcal{F}_n(\lambda_n) \end{bmatrix}. \tag{6b}$$

The problem in implementing the scheme (6a) then boils down to a fast computation of $\partial_\lambda F$.

## 3   Gradient computation

To compute $\partial_\lambda F$, we employ the integration by parts as described in [21, Appendix: The Gradient of the ELBO] for the Euclidean case, the proof can be trivially extended to the Riemannian setting via [7, Proposition 8.59]. The integration by parts yields the following expression for the gradient of the Free Energy

$$\partial_\lambda F\left[\prod_i q_{\lambda_i}(\theta_i), p\right](y) = \mathbb{E}_{\prod_i q_{\lambda_i}(\theta_i)}\left[\log \frac{\prod_i q_{\lambda_i}(\theta_i)}{p(y, \theta_1, \ldots, \theta_n)} \begin{bmatrix} T_1(\theta_1) - \partial_{\lambda_1} A(\lambda_1) \\ \ldots \\ T_n(\theta_n) - \partial_{\lambda_n} A(\lambda_n) \end{bmatrix}\right]. \tag{7}$$

The form of $\partial_\lambda F$ given in the equation (7) leads to an important observation: the differentiability of $F$ (both in the Euclidean and Riemannian senses) does not depend on the properties of $p(y, \theta_1, \ldots, \theta_n)$ with respect to $(\theta_1, \ldots, \theta_n)$. Instead, it depends solely on the existence of $\partial_{\lambda_i} A(\lambda_i)$. When $p(y, \theta_1, \ldots, \theta_n)$ is differentiable in terms of $(\theta_1, \ldots, \theta_n)$, a Monte Carlo method with favorable convergence properties is used to estimate the gradient (6a) with the reparameterization trick [16]. This approach can be extended to nondifferentiable functions by approximating the nondifferentiable function with a differentiable surrogate (smooth approximation), as in the RELAX method [10] and then applying the reparameterization trick.

The key observation of our study is as follows: we can construct a first-order approximation to $\partial_\lambda F$ using the form (7) without an explicit smooth approximation of $p(y, \theta_1, \ldots, \theta_n)$ via

$$\partial_\lambda F \approx \begin{bmatrix} \mathbb{E}_{q_{\lambda_1}(\theta_1)}\left[\log \frac{q_{\lambda_1}(\theta_1)}{p(y,\theta_1,\mu_2,\ldots,\mu_n)}(T_1(\theta_1) - \partial_{\lambda_1} A_1(\lambda_1))\right] \\ \vdots \\ \mathbb{E}_{q_{\lambda_n}(\theta_n)}\left[\log \frac{q_{\lambda_n}(\theta_n)}{p(y,\mu_1,\ldots,\theta_n)}(T_n(\theta_n) - \partial_{\lambda_n} A_n(\lambda_n))\right] \end{bmatrix}, \tag{8}$$

where $\mu_j = \mathbb{E}_{q_{\lambda_j}(\theta_j)}[\theta_j]$.

The error term for the approximation (8) can be obtained from Theorem 1 provided in Appendix E applied to each component of the gradient. For each $i$, we define a function $h_i$ that depends on all variables except $\theta_i$

$$h_{\theta_i}(\theta_1, \ldots, \theta_{i-1}, \theta_{i+1}, \ldots, \theta_n) = p(y, \theta_1, \ldots, \theta_{i-1}, \theta_i, \theta_{i+1}, \ldots, \theta_n) \tag{9}$$

Applying Theorem 1 to each $h_{\theta_i}$, we get

$$\mathbb{E}_{\prod_{j \neq i} q_{\lambda_j}(\theta_j)}[h_{\theta_i}(\theta_1, \ldots, \theta_{i-1}, \theta_{i+1}, \ldots, \theta_n)] \approx h_{\theta_i}(\mu_1, \ldots, \mu_{i-1}, \mu_{i+1}, \ldots, \mu_n), \tag{10}$$

where the approximation error is the limit expression from Theorem 1.

To estimate the right-hand side of equation (8) we employ REINFORCE estimator [25] corrected on the exponential family sufficient statistics (for details see Appendix C)

$$\Theta_i, \ldots, \Theta_N \sim q_{\lambda_i}(\theta_i) \tag{11a}$$

$$h_i(\lambda_i, \theta_i) = \log \frac{q_{\lambda_i}(\theta_i)}{p(y, \mu_1, \mu_2, \ldots, \theta_i, \ldots \mu_n)} (T_i(\theta_i) - \partial_{\lambda_i} A_i(\lambda_i)) \tag{11b}$$

$$f_i(\lambda_i, \theta_i) = T_i(\theta_i) - \partial_{\lambda_i} A(\lambda_i) \tag{11c}$$

$$A[h] = \frac{1}{N} \sum_i h_i(\lambda_i, \Theta_i) \tag{11d}$$

$$\text{Cov}[h, f] = \frac{1}{N} \sum_i (h_i(\lambda_i, \Theta_i) - A[h])^T f_i(\lambda_i, \Theta_i) \tag{11e}$$

$$\mathbb{E}_{q_{\lambda_i}(\theta_i)}[h_i(\lambda_i, \theta_i)] \approx \widetilde{\partial}_{\lambda_i} F = \frac{1}{N} \sum_i h_i(\lambda_i, \Theta_i) - \text{Cov}[h, f] \mathcal{F}_i(\lambda_i)^{-1} f_i(\lambda_i, \Theta_i). \tag{11f}$$

Algorithm 1 in Appendix D presents the complete implementation of the procedure described in Equation (6a). This algorithm integrates all key components of our method, offering a comprehensive description of RBBVI.

## 4 Experimental Results

Experiments presented in this section were implemented in Python [23] and Julia [6]. The code is publicly available at `https://github.com/biaslab/GradientFreeVI`. For the SIR model experiments in Subsection 4.1, we utilized the model implementation from Frost's SIR repository[1], integrating their Turing.jl [9] model specification. This integration allowed us to compare with the NUTS [12] and LSS (Latent Slice Sampling) [19] samplers using the Turing.jl framework.

### 4.1 Parameter Inference for a SIR Model

To evaluate our method, we use the Susceptible-Infected-Recovered (SIR) epidemiological model [4]. This provides a benchmark in an "ideal" scenario where gradients are readily available, allowing comparison against gradient-based and gradient-free approaches. The SIR model dynamics are governed by the following system of ordinary differential equations

$$\frac{dS}{dt} = -\beta c \frac{I}{N} S \qquad \frac{dI}{dt} = \beta c \frac{I}{N} S - \gamma I$$
$$\frac{dR}{dt} = \gamma I \qquad \frac{dC}{dt} = \beta c \frac{I}{N} S \tag{12}$$

where $S$, $I$, and $R$ represent the susceptible, infected, and recovered populations respectively, $C$ tracks cumulative cases, $N = S + I + R$ is the total population, $\beta$ is the infection rate, $c = 10$ is the contact rate, and $\gamma = 0.25$ is the recovery rate. We simulate daily infections by solving this system numerically with a population size of 1000 and observe infections through a Poisson distribution. Our task is to estimate the infection rate $\beta$ and the initial proportion of infected $i_0 = \frac{I(0)}{N}$ given only observed data, with uniform priors on both parameters.

We simulate daily infections using known parameters and model observed infections with a Poisson distribution. Our task is to estimate the infection rate $\beta$ and the initial proportion of infected $i_0$ given only observed data, with uniform priors. Our generative model is

$$i_0 \sim \text{Uniform}(0, 1) \qquad \beta \sim \text{Uniform}(0, 1)$$

$$C_t = \int_0^t \beta c \frac{I(\tau)}{N} S(\tau) d\tau \qquad X_t = C_t - C_{t-1} \qquad Y_t \sim \text{Poisson}(X_t). \tag{13}$$

For our approximate posterior, we use Beta distributions for $i_0$ and $\beta$, as they represent proportions in $[0, 1]$.

---

[1]`https://github.com/epirecipes/sir-julia`

We compared RBBVI with NUTS [12] and LSS samplers [19]. Table 1 (the table is provided in Appendix A) reveals distinct trade-offs between the methods. Under limited computational resources, our approach demonstrates strong performance, achieving superior coverage compared to nonparametric alternatives. The result highlights the key advantage of our parametric approach: efficient uncertainty quantification with restricted computational resources. However, as computational budgets increase, gradient-based methods like NUTS show their strengths through superior MSE values, while LSS also achieves excellent coverage. These results position our method as particularly valuable when computational efficiency is crucial or gradient information is unavailable. Meanwhile, we acknowledge that gradient-based methods like NUTS should be preferred when gradient information and substantial computational resources are available.

### 4.2 Inferring Neural Network training hyperparameter

In this experiment, we tune the learning rate for training a simple neural network on the MNIST dataset [18]. We cast this as an inverse problem, defining our forward generative model as training the network with a given learning rate $\varepsilon$

$$p(y, X, \varepsilon) = p(y|X, \varepsilon)p(\varepsilon)p(X)$$

with $p(X)$ uniform and

$$p(y|X, \varepsilon) \propto \exp(-L(y_v, f_\varepsilon(X_v))), \tag{14}$$

where

$$f_\varepsilon = f_{w(\varepsilon)}, \quad w(\varepsilon) = \underset{w}{\mathrm{argmin}} L(y_t, f_w(X_t)). \tag{15}$$

Here, $L$ is the loss function, $X_v, X_t, y_v, y_t$ are validation and training splits, and $f_w$ is the neural network with parameters $w$. The function $f_\varepsilon$ represents the trained neural network, where the optimal parameters $w(\varepsilon)$ depend on the learning rate $\varepsilon$ used during training. Crucially, each forward model evaluation requires training a network with the given learning rate.

We compare our method against several hyperparameter optimization techniques: Bayesian Optimization with Gaussian Processes (GP) [22] using different kernels: Radial Basis Function (RBF) and Matérn kernel [24] with $\nu = 1.0$ and $\nu = 2.5$, Tree-structured Parzen Estimator (TPE) [5]; and running inference over the same probabilistic model with LSS sampler [19].

The results in Table 2 (the table is provided in Appendix A) show that our method achieves comparable or better performance than advanced Bayesian optimization techniques with significantly fewer tuning steps. It provides direct uncertainty quantification for the optimal learning rate, enabling effective ensemble training. Our approach balances exploration and exploitation better than other methods, achieving high accuracy for individual models and ensembles. The detailed analysis of the experiment is provided in Appendix B.

## 5 Conclusion

In this work, we introduced RBBVI and demonstrated its application in two key areas: obtaining a posterior distribution over the unknown parameters of an ordinary differential equation (ODE) and tuning the learning rate of a neural network. Future research could investigate the assumption of a joint prior distribution over the parameters as mentioned in Equation 1. Additionally, our method could be utilized as a subroutine within larger probabilistic models. Notably, our algorithm was used as an inference subroutine in RxInfer.jl [3] in our experiments.

## Acknowledgments

This publication is part of the projects AUTO-AR and ROBUST (NWO: KICH3.LTP.20.006), which are partly financed by GN Hearing, the Eindhoven AI Systems Institute (EAISI), the Netherlands Enterprise Agency (RVO) and the Dutch Research Council (NWO).

We thank Simon Frost for suggesting the SIR model as an example to demonstrate our algorithm on the RxInfer discussion page[2].

---

[2]`https://github.com/orgs/ReactiveBayes/discussions/56`

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

# A Tables

| Method | $\beta$ Coverage | $i_0$ Coverage | MSE $\beta$ | MSE $i_0$ |
|---|---|---|---|---|
| * RBBVI | $0.21 \pm 0.13$ | $0.80 \pm 0.13$ | $0.45 \pm 0.03$ | $0.31 \pm 0.05$ |
| NUTS | $0.12 \pm 0.07$ | $0.0060 \pm 0.0027$ | $0.0011 \pm 0.0002$ | $0.18 \pm 0.06$ |
| LSS | $0.61 \pm 0.06$ | $0.013 \pm 0.007$ | $0.077 \pm 0.014$ | $0.36 \pm 0.05$ |
| ∘ RBBVI | $0.84 \pm 0.11$ | $0.50 \pm 0.14$ | $0.094 \pm 0.031$ | $0.17 \pm 0.00$ |
| NUTS | $0.87 \pm 0.02$ | $0.84 \pm 0.03$ | $0.00001 \pm 0.00000$ | $0.0067 \pm 0.0051$ |
| LSS | $0.99 \pm 0.00$ | $0.76 \pm 0.09$ | $0.0025 \pm 0.0019$ | $0.092 \pm 0.057$ |
| ⋆ RBBVI | $0.83 \pm 0.12$ | $0.80 \pm 0.13$ | $0.055 \pm 0.036$ | $0.064 \pm 0.040$ |
| NUTS | $0.94 \pm 0.02$ | $0.92 \pm 0.02$ | $0.00001 \pm 0.00000$ | $0.0054 \pm 0.0040$ |
| LSS | $0.98 \pm 0.01$ | $0.94 \pm 0.02$ | $0.0002 \pm 0.0002$ | $0.0089 \pm 0.0063$ |

**Table 1:** Coverage statistics and mean squared error (MSE) for inference in the SIR model. The symbols *, ∘, and ⋆ represent low, medium, and high computational budgets, respectively. These budgets are defined by the maximum number of calls to an ODE solver: up to 50,000 for low, 300,000 for medium, and 2,500,000 for high. Coverage is calculated as the proportion of times the true parameter value falls within the 95% Bayesian credible interval of the estimated posterior distribution. Higher coverage indicates that the method's uncertainty quantification is more reliable. Our RBBVI demonstrates particularly strong performance under low computational budgets, achieving superior coverage for both parameters compared to nonparametric alternatives and comparable MSE. This highlights RBBVI's ability to provide reliable uncertainty quantification even with limited computational resources. While NUTS shows better MSE due to its gradient-based nature, and LSS performs well with higher budgets, RBBVI's parametric approach offers a compelling advantage in resource-constrained scenarios. As computational budgets increase (∘ and ⋆), all methods show improved performance, with RBBVI maintaining competitive coverage and MSE values. This comparison demonstrates that our parametric method offers a robust alternative to nonparametric approaches, particularly excelling in scenarios where computational resources are limited while maintaining competitive performance at higher budgets.

| Method | Mean | Mode | Acc | Acc @ 50 | # Tuning steps |
|---|---|---|---|---|---|
| Gamma | 0.00291 | 0.00103 | 97.7% | 98.6% | 210 |
| Inverse Gamma | 0.00332 | 0.00111 | 97.3% | 98.7% | 330 |
| TPE | - | 0.00048 | 97.2% | - | 3000 |
| GP(RBF) | - | 0.00194 | 97.0% | - | 1000 |
| GP(Matern(1)) | - | 0.00418 | 97.3% | - | 1000 |
| GP(Matern(2.5)) | - | 0.00652 | 97.6% | - | 1000 |
| Cubature | 0.00332 | $\star$ | 97.7% | 98.6% | 210 |
| LSS | 0.0939 | $\star$ | 39.3% | 98.5% | 310 |

**Table 2:** Results of tuning the learning rate for a neural network trained on MNIST. The table shows the mean and mode of the posterior distribution over the learning rate, the test accuracy using the mode (Acc), the test accuracy of an ensemble of 50 models trained with learning rates sampled from the posterior (Acc @ 50), and the number of tuning steps. Our method (Gamma, Inverse Gamma) provides uncertainty estimates and competitive performance with fewer tuning steps than Bayesian optimization methods (TPE, GP variants). GP methods provide uncertainty estimates of validation loss but not of optimal learning rate. Further analysis is provided in Appendix B.

## B   Convergence of learning rate experiments

This section will elaborate on the number of tuning iterations needed in the experiments of subsection 4.2. To tune the learning rate, we train neural networks on a subset of the MNIST training dataset of size 3000. In addition to our method, we tuned the learning rate using Bayesian optimization with a Gaussian Process (GP) maximizer [22] and a tree-structured Parzen Estimator (TPE) [5]. Bayesian optimization methods do not have access to gradients. They are, therefore, prone to exploring low-probability regions of the search space, resulting in more iterations of training the neural network with candidate learning rates, which is very costly [8]. On the other hand, our method will search for a high-probability region of the search space and explore this region. The advantage is that we need fewer neural network training iterations to find a candidate learning rate, but the disadvantage is that we do not explore the entire search space. For every method, performance is evaluated on the validation dataset, a split of 1000 samples from the test dataset. The proposed learning rates are used in the Adam optimizer [16]. All experiments are run on a MacBook Pro 2021 M1 CPU, and none of the individual experiments take longer than 2 hours of CPU time.

### B.1   Tuning the learning rate with variational inference

With our method, we have to choose the functional form of the posterior distribution. To this extent, we choose three different distributions from the Exponential family: Exponential, Gamma, and Inverse Gamma Distributions. We run 100 iterations of variational inference for each distribution, each taking 30 samples. The total number of neural networks trained will be 3000 for every distribution. However, since we expect the Variational Free Energy to decrease and converge gradually, we employ an early stopping criterion that stops the tuning procedure when the difference between the Variational Free Energy of two iterations is less than 0.029. The choice for this stopping criterion was based on empirical results, and different values can be considered.

Since we know that the learning rate is typically a small number, we set the prior for every method to a distribution with the expected value at $\frac{1}{300}$. For the shape parameter of the Gamma distributions, we choose 1 as this corresponds to an Exponential distribution.

### B.2   Tuning the learning rate with Bayesian Optimization

We tune the learning rate with Bayesian optimization using a GP maximization procedure and a TPE. For the GP, we have to choose the kernel that is used. We run the tuning procedure with three different kernels: The Radial basis function (RBF) kernel and two variants of the Matern kernel with hyperparameters 1 and 2.5 [24]. The loss in the validation data set is optimized for these Bayesian optimization methods.

However, unlike our method, Bayesian optimization methods do not incrementally improve their estimate of the best learning rate. Instead, they explore a predefined search space to find the optimal learning rate. Therefore, the best learning rate seen so far in terms of the number of neural networks trained is a piecewise linear function with no guarantee of finding a better learning rate at a later stage. For this reason, employing a stopping criterion is impossible, as the best learning rate estimate is not adjusted incrementally.

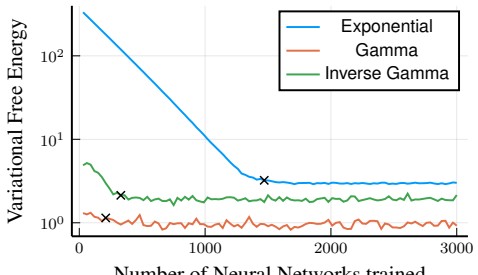 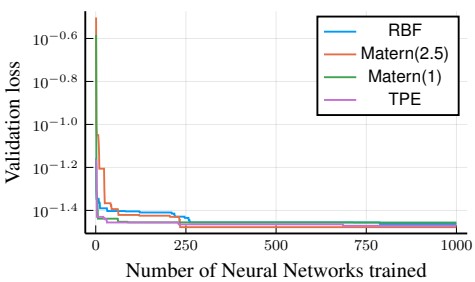

**(a)** Convergence plot of the Variational Free Energy using our method.

**(b)** Optimal validation loss found over the number of neural networks trained.

**Figure 1:** Convergence plots for the different methods. On the left-hand side, we see the convergence of the Variational Free Energy using our variational inference method. Here, the crosses indicate when our stopping criterion hit and which corresponding learning rate is used in subsection 4.2. On the right, we see the validation loss of the optimal learning rate found so far plotted over the number of neural networks trained. The optimal value so far is visualized because both the GP optimizers and the TPE explore the entire search space instead of incrementally improving their suggestions. This emphasizes that we cannot employ a stopping criterion for the convergence of these methods since some training curves remain constant for hundreds of iterations before finding a new optimal value. We use the optimal found learning rate for the experiments of subsection 4.2.

The GP maximization procedure explores the space of log-learning rates between $-7$ and $1$. This means a learning rate between $0.0009$ and $1$ is considered.

Maximizing the validation loss with a GP brings a significant disadvantage: fitting a GP has a computational cost of $\mathcal{O}(n^3)$, with $n$ being the number of samples. To iteratively fit a GP every time we train a new neural network and determine the optimal learning rate, therefore, it has a computational cost of $\mathcal{O}(n^4 + nc)$ with $n$ the number of tuning iterations and $c$ is the cost of training a neural network. This contrasts with the $\mathcal{O}(nc)$ run-time cost of variational inference. Because of this, we were able to run only 1000 iterations of the GP.

For the TPE, we describe the configuration space with a log-normal density with parameters $\mu = -5$ and $\sigma = 3$ .

### B.3 Experimental results

Since our method and the Bayesian Optimization methods optimize a different performance metric, we report them separately. In Figure 1, we see the curves of both methods. We see the Variational Free Energy over the number of neural networks trained, and we see the validation loss of the optimal learning rate found after $n$ neural networks trained. In addition, in Figure 1a, we can see the point at which our early stopping criterion triggers.

As we can see, our method gradually improves its suggestions for the optimal learning rate before converging. Although we cannot directly do gradient steps to improve our performance metric, this suggests that our method does not explore low-probability regions of the Bayesian posterior since the Variational Free Energy never spikes up to the value of the initial guess. In addition, because of this behavior, we can employ a stopping criterion in our method since we do not need to explore the entire search space.

## C Gradient estimator

We are interested in the estimation of the $\partial_{\lambda_i} F$ (8), rephrasing we are interested in the expectation of the following function

$$h_i(\lambda_i, \theta_i) = \log \frac{q_{\lambda_i}(\theta_i)}{p(y, \mu_1, \mu_2, \ldots, \theta_i, \ldots \mu_n)} (T_i(\theta_i) - \partial_{\lambda_i} A_i(\lambda_i)), \tag{16}$$

for that we employ the REINFORCE estimator [25] with a control variate . Specifically, we use the score function of the variational approximation as a control variate, which for exponential family distributions has the following form

$$f_i(\lambda_i, \theta_i) = \nabla_{\lambda_i} \log q_{\lambda_i}(\theta_i) = T_i(\theta_i) - \partial_{\lambda_i} A(\lambda_i). \tag{17}$$

This choice maintains the generic nature of our algorithm while allowing us to easily compute the expectation and the covariance matrix of the control variate

$$\mathbb{E}_{q_{\lambda_i}(\theta_i)}[f_i(\lambda_i, \theta_i)] = 0 \tag{18a}$$

$$\mathbb{E}_{q_{\lambda_i}(\theta_i)}[f_i(\lambda_i, \theta_i)^T f_i(\lambda_i, \theta_i)] = \mathcal{F}_i(\lambda_i) \tag{18b}$$

which results in the following estimator

$$\Theta_i, \ldots, \Theta_N \sim q_{\lambda_i}(\theta_i) \tag{19a}$$

$$A[h] = \frac{1}{N} \sum_i h_i(\lambda_i, \Theta_i) \tag{19b}$$

$$\text{Cov}[h, f] = \frac{1}{N} \sum_i (h_i(\lambda_i, \Theta_i) - A[h])^T f_i(\lambda_i, \Theta_i) \tag{19c}$$

$$\mathbb{E}_{q_{\lambda_i}(\theta_i)}[h_i(\lambda_i, \theta_i)] \approx \frac{1}{N} \sum_i h_i(\lambda_i, \Theta_i) - \text{Cov}[h, f]\mathcal{F}_i(\lambda_i)^{-1} f_i(\lambda_i, \Theta_i). \tag{19d}$$

## D    Main Algorithm

Our RBBVI approximates posterior distributions without requiring gradients of the log-probability density function. The algorithm implements an iterative optimization procedure that leverages Riemannian geometry and a gradient-free estimator based on the REINFORCE estimator [25] with a control variate. This approach enables optimization even for nondifferentiable forward models. Algorithm 1 provides a detailed implementation, synthesizing all key components into a comprehensive approach. For conciseness, we denote the Riemannian metrics on each $\Lambda_i$ as

$$\langle v, w \rangle_{\lambda_i}^i = v^T \mathcal{F}_i(\lambda_i) w. \tag{20}$$

## E    Taylor Expansion of the Smooth Approximation

Before formally stating the main result, let us introduce a technique for smoothing continuous functions, which can be particularly useful for nondifferentiable processes. Consider a continuous function $h : \mathbb{R}^n \to \mathbb{R}$. We can define a smoothed version of this function as follows

$$\widetilde{h}_\sigma(x) = \mathbb{E}_{\mathcal{N}(x, \sigma\mathbb{I})}[h(x^*)] \tag{21}$$

where $\sigma > 0$ is a smoothing parameter. This new function $\widetilde{h}_\sigma$ is differentiable even if the original function $h$ is not. Moreover, as $\sigma \to 0$, $\widetilde{h}_\sigma$ converges to $h$.

To illustrate this technique, let us consider the Wiener process (also known as Brownian motion) as an example of a continuous but nowhere differentiable function. Let $\{W(t)\}_{t \geq 0}$ be a standard Wiener process defined in a probability space $(\Omega, \mathcal{F}, \mathbb{P})$. Although we will not explore its full properties, it is worth noting that $W(t)$ is almost surely continuous but not differentiable at any point.

Now, let us apply our smoothing technique to a sample path of the Wiener process. Figure 2 shows a realization of $W(t)$ in the interval $[-1, 1]$, together with three smoothed approximations $\widetilde{W}_\sigma(t)$ for different values of $\sigma$. In this figure, we observe the original Wiener process $W(t)$ (black line), which is highly irregular and nondifferentiable, alongside three smoothed approximations $\widetilde{W}\sigma(t)$ for decreasing values of $\sigma$. The green line ($\sigma = 0.707$) shows a very smooth approximation that captures the general trend but misses fine details, the red line ($\sigma = 0.224$) presents a moderately smooth approximation that captures more detail while remaining differentiable, and the blue line ($\sigma = 0.095$) offers a closer approximation that nearly overlaps with the original process while maintaining differentiability. As $\sigma$ decreases, we observe that $\widetilde{W}_\sigma(t)$ approaches $W(t)$ more closely, illustrating how our smoothing technique can approximate nondifferentiable functions with differentiable ones. This example provides intuition for the more general result we are about to present, which extends this idea to expectations over arbitrary continuous functions.

Let $f : \mathbb{R}^n \to \mathbb{R}$ be a smooth function and $q$ be a continuous probability distribution with support on an open subset $U \subseteq \mathbb{R}^n$. We are interested in computing $\mathbb{E}_{q(x)}[f(x)]$. Using a Taylor expansion

---

**Algorithm 1** Riemannian Black Box Variational Inference

---

**Input:** Initial priors $p_i(\theta_i)$, observed data $y$, forward model $F$, maximum iterations $N$, approximate families $Q_i$, Riemannian exponential family manifolds $*_i$, retractions $R_i$ from $T_{*_\rangle}$ to $\Lambda_i$, step size sequence $\alpha_k$, initial points $\lambda_i \in \Lambda_i$, gradient tolerances $\varepsilon_k$, running average window $W$, overall tolerance $\varepsilon$

**Output:** Approximate posterior distributions $q(\theta_i)$ for $i = 1, \ldots, n$

1: Initialize $q^0(\theta_i) = p_i(\theta_i)$ for $i = 1, \ldots, n$
2: **for** $t = 1$ to $N$ **do**
3:     **for** $i = 1$ to $n$ **do**
4:         $\mu_i \leftarrow \mathbb{E}_{q^{t-1}(\theta_i)}[\theta_i]$
5:     **end for**
6:     **for** $i = 1$ to $n$ **do**
7:         $\lambda_0 \leftarrow$ parameters of $q^{t-1}(\theta_i)$
8:         **for** $k = 0, 1, 2, \ldots$ **do**
9:             **for** $s = 1$ to $S$ **do**
10:                $\Theta[s] \sim q_{\lambda_k}(\theta)$
11:                $h_i[s] = T_i(\Theta[s]) - \partial_{\lambda_i} A_i(\lambda_i)$
12:                $f_i[s] = (\log q_{\lambda_i}(\Theta[s]) - \log p_i(y, \mu_1, \ldots, \Theta[s], \ldots, \mu_n)) h_i[s]$
13:             **end for**
14:             Estimate $\text{Cov}[h, f]$ (11e)
15:             Estimate $\widetilde{\partial}_{\lambda_k} F$ (11f)
16:             Update: $\lambda_{k+1} = R_{\lambda_k}(-\alpha_k \widetilde{\partial}_{\lambda_k} F)$
17:             **if** $\langle \widetilde{\partial}_{\lambda_k} F, \widetilde{\partial}_{\lambda_k} F \rangle^i_{\lambda_i} \leqslant \varepsilon_i$ (20) **then**
18:                **break**
19:             **end if**
20:         **end for**
21:         $q^t(\theta_i) = h_i(\theta_i) \exp(T_i(\theta_i)^\top \lambda_i^{k+1} - A(\lambda_i^{k+1}))$
22:     **end for**
23:     **for** $i = 1$ to $n$ **do**
24:         $\mu_i \leftarrow \mathbb{E}_{q^t(\theta_i)}[\theta_i]$
25:     **end for**
26:     $L[i] = \log p(y, \mu_1, \ldots, \mu_n)$
27:     **if** Running Average Improvement with window $W$ of $L < \varepsilon$ **then**
28:         **break**
29:     **end if**
30: **end for**
31: **return** $q^N(\theta_i)$ for $i = 1, \ldots, n$

---

with Residual Term, we know that for any point $x^*$ in which one $f$ is analytic, exists such $\hat{x} \in \mathbb{R}^n$ that the following property identity holds

$$f(x) = f(x^*) + \nabla_x f(x)|_{x=x^*}(x - x^*) + (x - x^*)^T H(\hat{x})(x - x^*). \tag{22}$$

Using the fact (22) assuming that $f$ is analytic in $\mu = \mathbb{E}_{q(x)}[x]$, we obtain the following identity

$$\mathbb{E}_{q(x)}[f(x)] = f(\mu) + \mathbb{E}_{q(x)}\left[(x - \mu)^T H(\hat{x})(x - \mu)\right], \tag{23}$$

it means that first order approximation to the expectation $\mathbb{E}_{q(x)}[f(x)]$ is $f(\mu)$.

**Theorem 1.** *Let*

$$h : \mathbb{R}^n \to \mathbb{R} \tag{24}$$

*be a continuous function and q be a continuous distribution over an open set $U \subseteq \mathbb{R}^n$. Consider the following function*

$$\widetilde{h}_\sigma(x) = \mathbb{E}_{x^* \sim \mathcal{N}(x, \sigma\mathbb{I})}[h(x^*)]. \tag{25}$$

*Then for $\mu = \mathbb{E}_q[x]$ there exists a $\tilde{x} \in U$ such that the following identity holds*

$$\mathbb{E}_q[h(x)] = h(\mu) + \lim_{\sigma \to 0} \mathbb{E}_q\left[(x - \tilde{x})^T \frac{\nabla^2_{x^*} \widetilde{h}_\sigma(x^*)|_{x^*=\tilde{x}}}{2}(x - \tilde{x})\right]. \tag{26}$$

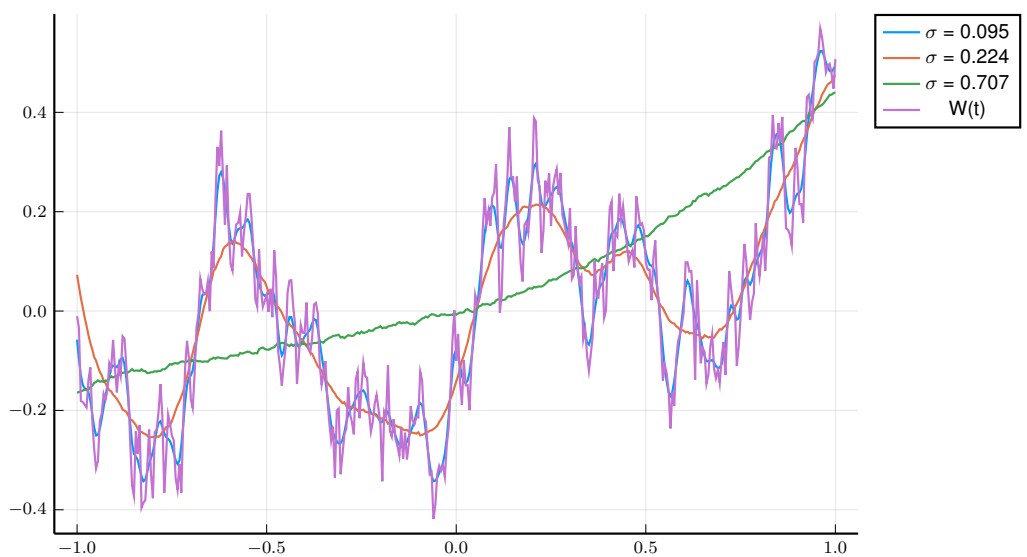

**Figure 2:** A sample path of the Wiener process and its smoothed approximations

*Proof.* Let $h : \mathbb{R}^n \to \mathbb{R}$ be a continuous function and $q$ be a continuous distribution over an open set $U \subseteq \mathbb{R}^n$. Define $\widetilde{h}_\sigma(x) = \mathbb{E}_{x^* \sim \mathcal{N}(x, \sigma \mathbb{I})}[h(x^*)]$ and let $\mu = \mathbb{E}_q[x]$.

First, note that $\widetilde{h}_\sigma(x)$ is infinitely differentiable for all $\sigma > 0$.

By the definition of expectation and the properties of $\widetilde{h}_\sigma(x)$, we have

$$\mathbb{E}_q[h(x)] = \lim_{\sigma \to 0} \mathbb{E}_q[\widetilde{h}_\sigma(x)]$$

For any fixed $\sigma > 0$, we can apply Taylor's theorem to $\widetilde{h}_\sigma(x)$ around $\mu$. There exists a point $\tilde{x}_\sigma$ on the line segment between $x$ and $\mu$ such that

$$\widetilde{h}_\sigma(x) = \widetilde{h}_\sigma(\mu) + \nabla \widetilde{h}_\sigma(\mu)^T (x - \mu) + \frac{1}{2}(x - \mu)^T \nabla^2 \widetilde{h}_\sigma(\tilde{x}_\sigma)(x - \mu)$$

Taking the expectation with respect to $q(x)$ on both sides

$$\mathbb{E}_q\left[\widetilde{h}_\sigma(x)\right] = \widetilde{h}_\sigma(\mu) + \mathbb{E}_q\left[\nabla \widetilde{h}_\sigma(\mu)^T (x - \mu)\right] + \frac{1}{2}\mathbb{E}_q[(x - \mu)^T \nabla^2 \widetilde{h}_\sigma(\tilde{x}_\sigma)(x - \mu)]$$

Note that $\mathbb{E}_q[(x - \mu)] = 0$, so the second term on the right-hand side vanishes.

Now, let $\sigma \to 0$. We know that $\lim_{\sigma \to 0} \widetilde{h}_\sigma(x) = h(x)$ pointwise, so

$$\lim_{\sigma \to 0} \widetilde{h}_\sigma(\mu) = h(\mu)$$

By the continuity of $h$ and the boundedness of the Gaussian kernel, we can apply the dominated convergence theorem to swap the limit and the expectation

$$\lim_{\sigma \to 0} \mathbb{E}_q\left[\widetilde{h}_\sigma(x)\right] = \mathbb{E}_q[h(x)].$$

For the Hessian term, there exists a subsequence $\sigma_k \to 0$ such that $\tilde{x}_{\sigma_k}$ converges to some $\tilde{x} \in U$. This is because $U$ is open and contains the line segment between $x$ and $\mu$.

Combining all these results, we obtain the statement of the theorem

$$\mathbb{E}_q[h(x)] = h(\mu) + \lim_{\sigma \to 0} \frac{1}{2}\mathbb{E}_q[(x - \mu)^T \nabla^2 \widetilde{h}_\sigma(\tilde{x}_\sigma)(x - \mu)].$$

$\square$

