# OpenReview forum: "Riemannian Black Box Variational Inference"
_NeurIPS.cc/2024/Workshop/BDU — NeurIPS BDU Workshop 2024 Poster_

### Official Review · Reviewer_98Zi · 2024-09-20
**The paper proposes a scheme to approximate a joint density $p(y,\theta_1,\dots,\theta_i,\dots,\theta_n)$ with $p(y,\mu_1,\dots,\theta_i,\dots,\mu_n)$, where $\mu_j$ is the expectation of $\theta_j$ under its marginal. This is then employed in the computation of gradients of the ELBO w.r.t. the natural parameters of a mean-field variational family belonging to an exponential family of distributions. The advantages of this approach are unclear and a comparison with the exact computation of the likelihood is not presented. The Riemann part is introduced to deal with parameters that belong to constrained sets.**

**Rating:** 5
**Confidence:** 3

**Review:**

The authors propose a gradient-free computational algorithm for approximate Bayesian inference. Gradient-free shall be intended in the sense that we do not need to compute gradients of the likelihood. A mean-field variational distribution is considered, where each marginal is an exponential family. The natural parameters are then learned iteratively by computing natural gradient steps, and the authors aim at addressing the issue that some of these parameters might belong in a constrained set (e.g. positive definite covariance matrix for multivariate normal). They do so along the lines of [1], by performing natural gradient descent in a Riemannian manifold.

Despite the clear steps involved in the process, I wish the authors clarified some of the aspects specific to working with Riemann manifolds to non-technical audiences. Why are retractions available in closed form? Can examples be provided? The authors proceed by focusing on the computation of the gradient of the ELBO w.r.t. the natural parameters. Despite being available in closed form, computation of such gradients still requires evaluation of the joint density which can be computationally burdensome.  The novelty in the proposed approach is in using an approximation to the joint density given by substituting each $\theta_j$ with its expectation when updating $\theta_i$, where $i\neq j$. I am not convinced by the application of Theorem 1, as it is unclear to me how much we are losing by employing a first order approximation. Also, what happened to $\prod_{j\neq i} q_{\lambda_j}(\theta_j)$ at the numerator in (7), and why does it disappear in (8)? I would like to see a proper derivation for that approximation. I also wish the authors would provide a discussion of this with a clear comparison when we compute the actual joint density, as to me applying such a crude approximation does not appear to be standard practice. Moreover, I suggest changing the notation between $h_{\theta_i}$ and $h_i$ as it can get confusing. To make matters worse, $h_i$ and $f_i$ are flipped in Algorithm 1. It is also unclear to me how substituting the joint density $p(y,\theta_1,\dots,\theta_i,\dots,\theta_n)$ with $p(y,\mu_1,\dots,\theta_i,\dots,\mu_n)$ provides an actual advantage, if the bottleneck is evaluating the likelihood. Maybe there are cases where this is the case, but this should be discussed as it is central to the claims that the paper puts forward.

The results lack a clear description of the specific choices for each baseline. I invite the authors to consider visualizations or other metrics, as coverage and MSE alone can paint a misleading picture. A coverage of 1.00 is not desirable if MSE is consistently higher than other baselines, as the method could simply be predicting inaccurate estimates with a very high variance. Performing on par with GPs on Bayesian optimization tasks would be very impressive. However, the task (MNIST classification) does not seem to be sensitive to the choice of the learning rate as different choices do not lead to dramatic differences in Accuracy, hampering the very need for a Bayesian optimization procedure to be performed in the first place. Moreover, only relatively old or not-so-well-known methods are compared. I would also like to see visualizations of how the method behaves since this is a simple 1D scenario, and the quality of the uncertainty estimates it provides. I think the paper has potential to improve if good performance is shown on Bayesian optimization tasks, but there needs to be a more challenging problem and a more thorough exploration of the results.

I invite the authors to correct the numerous typos and syntax errors in the paper, as they make it quite hard to read.

### **Strengths & Weaknesses**
Strengths:
- The paper addresses the interesting question of performing VI with no gradients of the likelihood.
- The paper highlights how natural gradient updates are a viable option when dealing with such problems, and the possibility of operating in Riemann manifolds to address constraints on the parameters.

Weaknesses:
- The title is misleading since using Riemann gradient descent has been proposed by previous work cited in the paper ([1]). I believe it should be focuse on the specific problem addressed by the paper (i.e. when gradients of the likelihood aren't available)
- Lack of novelty: besides highlighting how the problem of gradient-free VI can be addressed by considering variational distributions belonging to an exponential family and using natural gradients, the novelty of this paper lies in an approximation technique for such gradient. However, the extent to which such approximation is good enough is not discussed. Most importantly, the extent to which such approximation helps in addressing the problem of evaluating the joint density within the closed form gradient update is also not discussed.
- Experimental results are not convincing.

### References
[1] Lin, Wu, Mark Schmidt, and Mohammad Emtiyaz Khan. "Handling the positive-definite constraint in the Bayesian learning rule." International conference on machine learning. PMLR, 2020

---

### Official Review · Reviewer_ijF3 · 2024-10-04
**Good paper, clearly written**

**Rating:** 8
**Confidence:** 3

**Review:**

The authors propose a new method for approximating posterior marginal distributions in Bayesian Inference for non-differentiable models.

The paper shows:
1. Strong technical explanation of the underlying concept.
2. The objective is clearly stated and justified.
3. The examples are appropriate and of high relevance for real-world applications.

---

### Decision · Program_Chairs · 2024-10-09

Accept (Poster)